# Segregated Graphs and Marginals of Chain Graph Models

**Ilya Shpitser**
Department of Computer Science
Johns Hopkins University
ilyas@cs.jhu.edu

## Abstract

Bayesian networks are a popular representation of asymmetric (for example causal) relationships between random variables. Markov random fields (MRFs) are a complementary model of symmetric relationships used in computer vision, spatial modeling, and social and gene expression networks. A chain graph model under the Lauritzen-Wermuth-Frydenberg interpretation (hereafter a chain graph model) generalizes both Bayesian networks and MRFs, and can represent asymmetric and symmetric relationships together.

As in other graphical models, the set of marginals from distributions in a chain graph model induced by the presence of hidden variables forms a complex model. One recent approach to the study of marginal graphical models is to consider a well-behaved supermodel. Such a supermodel of marginals of Bayesian networks, defined only by conditional independences, and termed the *ordinary Markov model*, was studied at length in [6].

In this paper, we show that special mixed graphs which we call *segregated graphs* can be associated, via a Markov property, with supermodels of marginals of chain graphs defined only by conditional independences. Special features of segregated graphs imply the existence of a very natural factorization for these supermodels, and imply many existing results on the chain graph model, and the ordinary Markov model carry over. Our results suggest that segregated graphs define an analogue of the ordinary Markov model for marginals of chain graph models.

We illustrate the utility of segregated graphs for analyzing outcome interference in causal inference via simulated datasets.

## 1 Introduction

Graphical models are a flexible and widely used tool for modeling and inference in high dimensional settings. Directed acyclic graph (DAG) models, also known as Bayesian networks [11, 8], are often used to model relationships with an inherent asymmetry, perhaps induced by a temporal order on variables, or cause-effect relationships. Models represented by undirected graphs (UGs), such as Markov random fields (MRFs), are used to model symmetric relationships, for instance proximity in social graphs, expression co-occurrence in gene networks, coinciding magnetization of neighboring atoms, or similar colors of neighboring pixels in an image.

Some graphical models can represent both symmetric and asymmetric relationships together. One such model is the chain graph model under the Lauritzen-Wermuth-Frydenberg interpretation, which we will shorten to "the chain graph model." We will not consider the chain graph model under the Andersen-Madigan-Perlman (AMP) interpretation, or other chain graph models [22, 1] discussed in [4] in this paper. Just as the DAG models and MRFs, the chain graph model has a set of equivalent (under some assumptions) definitions via a set of Markov properties, and a factorization.

Modeling and inference in multivariate settings is complicated by the presence of hidden, yet relevant variables. Their presence motivates the study of marginal graphical models. Marginal DAG models are complicated objects, inducing not only conditional independence constraints, but also more general equality constraints such as the "Verma constraint" [21], and inequality constraints such as the instrumental variable inequality [3], and the Bell inequality in quantum mechanics [2].

One approach to studying marginal DAG models has therefore been to consider tractable supermodels defined by some easily characterized set of constraints, and represented by a mixed graph. One such supermodel, defined only by conditional independence constraints induced by the underlying hidden variable DAG on the observed margin, is the ordinary Markov model, studied in depth in [6]. Another supermodel, defined by generalized independence constraints including the Verma constraint [21] as a special case, is the nested Markov model [16]. There is a rich literature on Markov properties of mixed graphs, and corresponding independence models. See for instance [15, 14, 7].

In this paper, we adapt a similar approach to the study of marginal chain graph models. Specifically, we consider a supermodel defined only by conditional independences on observed variables of a hidden variable chain graph, and ignore generalized equality constraints and inequalities. We show that we can associate this supermodel with special mixed graphs which we call *segregated graphs* via a global Markov property. Special features of segregated graphs imply the existence of a convenient factorization, which we show is equivalent to the Markov property for positive distributions. This equivalence, along with properties of the factorization, implies many existing results on the chain graph model, and the ordinary Markov model carry over.

The paper is organized as follows. Section 2 describes a motivating example from causal inference for the use of hidden variable chain graphs, with details deferred until section 6. In section 3, we introduce the necessary background on graphs and probability theory, define segregated graphs (SGs) and an associated global Markov property, and show that the global Markov properties for DAG models, chain graph models, and the ordinary Markov model induced by hidden variable DAGs are special cases. In section 4, we define the model of conditional independence induced by hidden variable chain graphs, and show it can always be represented by a SG via an appropriate global Markov property. In section 5, we define *segregated factorization* and show that under positivity, the global Markov property in section 4 and segregated factorization are equivalent. In section 6, we introduce causal inference and interference analysis as an application domain for hidden variable chain graph models, and thus for SGs, and discuss a simulation study that illustrates our results and shows how parameters of the model represented by a SG can directly encode parameters representing outcome interference in the underlying hidden variable chain graph. Section 7 contains our conclusions. We will provide outlines of arguments for our claims below, but will generally defer detailed proofs to the supplementary material.

## 2 Motivating Example: Interference in Causal Inference

Consider a dataset obtained from a placebo-controlled vaccination trial, described in [20], consisting of mother/child pairs where the children were vaccinated against pertussis. We suspect that though mothers were not vaccinated directly, the fact that children were vaccinated, and each mother will generally only contract pertussis from her child, the child's vaccine may have a protective effect on the mother. At the same time, if only the mothers but not children were vaccinated, we would expect the same protective effect to operate in reverse. This is an example of *interference*, an effect of treatment on experimental units other than those to which the treatment was administered. The relationship between the outcomes of mother and child due to interference in this case has some features of a causal relationship, but is symmetric.

We model this study by a chain graph shown in Fig. 1 (a), see section 6 for a justification of this model. Here $B_1$ is the vaccine (or placebo) given to children, and $Y_1$ is the children's outcomes. $B_2$ is the treatment given to mothers (in our case no treatment), and $Y_2$ is the mothers' outcomes. Directed edges represent the direct causal effect of treatment on unit, and the undirected edge represents the interference relationship among the mother/child outcome pair. In this model $(B_1 \perp\!\!\!\perp B_2)$ (mother and child treatment are assigned independently), and $(Y_1 \perp\!\!\!\perp B_2 \mid B_1, Y_1)$, $(Y_2 \perp\!\!\!\perp B_1 \mid B_2, Y_1)$ (mother's outcome is independent of child's treatment, if we know child's outcome, and mother's treatment, and vice versa). Since treatments in this study were randomly assigned, there are no unobserved confounders.

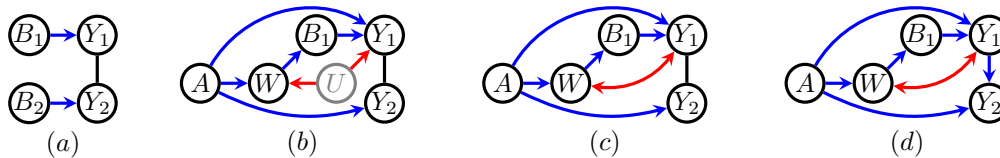

Figure 1: (a) A chain graph representing the mother/child vaccination example in [20]. (b) A more complex vaccination example with a followup booster shot. (c) A naive generalization of the latent projection idea applied to (b), where $\leftrightarrow$ and $-$ edges meet. (d) A segregated graph preserving conditional independences in (b) not involving $U$.

Consider, however, a more complex example, where both mother and child are given the initial vaccine $A$, but possibly based on results $W$ of a followup visit, children are given a booster $B_1$, and we consider the child's ($Y_1$) and the mother's ($Y_2$) outcomes, where the same kind of interference relationship is operating. We model the child's unobserved health status, which influences both $W$ and $Y_1$, by a (possibly very high dimensional) hidden variable $U$. The result is a hidden variable chain graph in Fig. 1 (b). Since $U$ is unobserved and possibly very complex, modeling it directly may lead to model misspecification. An alternative, explored for instance in [13, 6, 16], is to consider a model defined by conditional independences induced by the hidden variable model in Fig. 1 (b) on observed variables $A, B_1, W, Y_1, Y_2$.

A simple approach that directly generalizes what had been done in DAG models is to encode conditional independences via a path separation criterion on a mixed graph constructed from a hidden variable chain graph via a *latent projection* operation [21]. The difficulty with this approach is that simple generalizations of latent projections to the chain graph case may yield graphs where $\leftrightarrow$ and $-$ edges met, as happens in Fig. 1 (c). This is an undesirable feature of a graphical representation, since existing factorization and parameterization results for chain graphs or ordinary Markov models, which decompose the joint distribution into pieces corresponding to sets connected by $-$ or $\leftrightarrow$ edges, do not generalize.

In the remainder of the paper, we show that for any hidden variable chain graph it is always possible to construct a (not necessarily unique) mixed graph called a *segregated graph (SG)* where $\leftrightarrow$ and $-$ edges do not meet, and which preserves all conditional independences on the observed variables. One SG for our example is shown in Fig. 1 (d). Conditional independences implied by this graph are $B_1 \perp\!\!\!\perp A_1 \mid W$ and $Y_2 \perp\!\!\!\perp W, B_1 \mid A_1, Y_1$. Properties of SGs imply existing results on chain graphs and the ordinary Markov model carry over with little change. For example, we may directly apply the parameterization in [6], and the fitting algorithm in [5] to the model corresponding to Fig. 1 (d) if the state spaces are discrete, as we illustrate in section 6.1. The construction we give for SGs may replace undirected edges by directed edges in a way that may break the symmetry of the underlying interference relationship. Thus, directed edges in a SG do not have a straightforward causal interpretation.

## 3 Background and Preliminaries

We will consider mixed graphs with three types of edges, undirected ($-$), directed ($\leftrightarrow$), and directed ($\rightarrow$), where a pair of vertices is connected either by a single edge, or a pair of edges one of which is directed and one bidirected. We will denote an edge as an ordered pair of vertices with a subscript indicating the type of edge, for example $(AB)_\rightarrow$. We will suppress the subscript if edge orientation is not important. An alternating sequence of nodes and edges of the form $A_1, (A_1 A_2), A_2, (A_2 A_3), A_3, \ldots A_{i-1}, (A_{i-1} A_i), A_i$ where we allow $A_i = A_j$ if $i \neq j \pm 1$ is called a *walk* (in some references also a *route*). We will denote walks by lowercase Greek letters. A walk with non-repeating edges is called a *trail*. A trial with non-repeating vertices is called a *path*. A *directed cycle* is a trail of the form $A_1, (A_1 A_2)_\rightarrow, A_2, \ldots, A_i, (A_i A_1)_\rightarrow, A_1$. A *partially directed cycle* is a trail with $-, \rightarrow$ edges, and at least one $\rightarrow$ edge where there exists a way to orient $-$ edges to create a directed cycle. We will sometimes write a path from $A$ to $B$ where intermediate vertices are not important, but edge orientation is as, for example, $A \rightarrow \circ - \ldots - \circ - B$.

A mixed graph with no $-$ and $\leftrightarrow$ edges, and no directed cycles is called a directed acyclic graph (DAG). A mixed graph with no $-$ edges, and no directed cycles is called an acyclic directed mixed graph (ADMG). A mixed graph with no $\leftrightarrow$ edges, and no partially directed cycles is called a chain graph (CG). A *segregated graph* (SG) is a mixed graph with no partially directed cycles where no path of the form $A_i(A_iA_j)_-A_j(A_jA_k)_{\leftrightarrow}A_k$ exists. DAGs are special cases of ADMGs and CGs which are special cases of SGs.

We consider sets of distributions over a set $\mathbf{V}$ defined by independence constraints linked to above types of graphs via (global) Markov properties. We will refer to $\mathbf{V}$ as either vertices in a graph or random variables in a distribution, it will be clear from context what we mean.

A Markov model of a graph $\mathcal{G}$ defined via a global Markov property has the general form

$$\mathcal{P}(\mathcal{G}) \equiv \left\{ p(\mathbf{V}) \Big| (\forall \mathbf{A} \dot\cup \mathbf{B} \dot\cup \mathbf{C} \subseteq \mathbf{V}), (\mathbf{A} \perp\!\!\!\perp \mathbf{B} \mid \mathbf{C})_{\mathcal{G}} \Rightarrow (\mathbf{A} \perp\!\!\!\perp \mathbf{B} \mid \mathbf{C})_{p(\mathbf{V})} \right\},$$

where the consequent means "$\mathbf{A}$ is independent of $\mathbf{B}$ conditional on $\mathbf{C}$ in $p(\mathbf{V})$," and the antecedent means "$\mathbf{A}$ is separated from $\mathbf{B}$ given $\mathbf{C}$ according to a certain walk separation property in $\mathcal{G}$." Since DAGs, ADMGs, and CGs are special cases of SGs, we will define the appropriate path separation property for SGs, which will recover known separation properties in DAGs, ADMGs and CGs as special cases.

A walk $\alpha$ contained in another walk $\beta$ is called a subwalk of $\beta$. A maximal subwalk in $\beta$ where all edges are undirected is called a *section* of $\beta$. A section may consist of a single node and no edges. We say a section $\alpha$ of a walk $\beta$ is a *collider section* if edges in $\beta$ immediately preceding and following $\alpha$ contain arrowheads into $\alpha$. Otherwise, $\alpha$ is a non-collider section. A walk $\beta$ from $A$ to $B$ is said to be *s-separated* by a set $\mathbf{C}$ in a SG $\mathcal{G}$ if there exists a collider section $\alpha$ that does not contain an element of $\mathbf{C}$, or a non-collider section that does (such a section is called blocked). $\mathbf{A}$ is said to be *s-separated* from $\mathbf{B}$ given $\mathbf{C}$ in a SG $\mathcal{G}$ if every walk from a vertex in $\mathbf{A}$ to a vertex in $\mathbf{B}$ is s-separated by $\mathbf{C}$, and is *s-connected* given $\mathbf{C}$ otherwise.

**Lemma 3.1** *The Markov properties defined by superactive routes (walks) [17] in CGs, m-separation [14] in ADMGs, and d-separation [11] in DAGs are special cases of the Markov property defined by s-separation in SGs.*

## 4  A Segregated Graph Representation of CG Independence Models

For a SG $\mathcal{G}$, and $\mathbf{W} \subset \mathbf{V}$, define the model $\mathcal{P}(\mathcal{G})^{\mathbf{W}}$ to be the set of distributions where all conditional independences in $\sum_{\mathbf{W}} p(\mathbf{V})$ implied by $\mathcal{G}$ hold. That is

$$\mathcal{P}(\mathcal{G})^{\mathbf{W}} \equiv \left\{ p(\mathbf{V} \setminus \mathbf{W}) \Big| (\forall \mathbf{A} \dot\cup \mathbf{B} \dot\cup \mathbf{C} \subseteq \mathbf{V} \setminus \mathbf{W}), (\mathbf{A} \perp\!\!\!\perp \mathbf{B} \mid \mathbf{C})_{\mathcal{G}} \Rightarrow (\mathbf{A} \perp\!\!\!\perp \mathbf{B} \mid \mathbf{C})_{p(\mathbf{V})} \right\}.$$

$\mathcal{P}(\mathcal{G}_1)^{\mathbf{W}_1}$ may equal $\mathcal{P}(\mathcal{G}_2)^{\mathbf{W}_2}$ even if $\mathcal{G}_1, \mathbf{W}_1$ and $\mathcal{G}_2, \mathbf{W}_2$ are distinct. If $\mathbf{W}$ is empty, $\mathcal{P}(\mathcal{G})^{\mathbf{W}}$ simply reduces to the Markov model defined by s-separation on the entire graph.

We are going to show that there is always a SG that represents the conditional independences that define $\mathcal{P}(\mathcal{G})^{\mathbf{W}}$, using a special type of vertex we call *sensitive*. A vertex $V$ in an SG $\mathcal{G}$ is *sensitive* if for any other vertex $W$, if $W \to \circ - \ldots - \circ - V$ exists in $\mathcal{G}$, then $W \to V$ exists in $\mathcal{G}$. We first show that if $V$ is sensitive, we can orient all undirected edges away from $V$ and this results in a new SG that gives the same set of conditional independence via s-separation. This is Lemma 4.1. Next, we show that for any $V$ with a child $Z$ with adjacent undirected edges, if $Z$ is not sensitive, we can make it sensitive by adding appropriate edges, and this results in a new SG that preserves all conditional independences that do not involve $V$. This is Lemma 4.3. Given above, for any vertex $V$ in a SG $\mathcal{G}$, we can construct a new SG that preserves all conditional independences in $\mathcal{G}$ that do not involve $V$, and where no children of $V$ have adjacent undirected edges. This is Lemma 4.4. We then "project out $V$" to get another SG that preserves all conditional independences not involving $V$ in $\mathcal{G}$. This is Theorem 4.1. We are then done, Corollary 4.1 states that there is always a (not necessarily unique) SG for the conditional independence structure of a marginal of a CG.

**Lemma 4.1** *For $V$ sensitive in a SG $\mathcal{G}$, let $\mathcal{G}^{\langle V \rangle}$ be the graph be obtained from $\mathcal{G}$ by replacing all $-$ edges adjacent to $V$ by $\to$ edges pointing away from $V$. Then $\mathcal{G}^{\langle V \rangle}$ is an SG, and $\mathcal{P}(\mathcal{G}) = \mathcal{P}(\mathcal{G}^{\langle V \rangle})$.*

The intuition here is that directed edges differ from undirected edges due to collider bias induced by the former. That is, dependence between parents of a block is created by conditioning on variables in the block. But a sensitive vertex in a block is already dependent on all the parents in the block, so orienting undirected edges away from such a vertex and making it a block parent does not change the set of advertised independences.

**Lemma 4.2** *Let $\mathcal{G}$ be an SG, and $\mathcal{G}'$ a graph obtained from adding an edge $W \to V$ for two non-adjacent vertices $W, V$ where $W \to \circ - \ldots - \circ - V$ exists in $\mathcal{G}$. Then $\mathcal{G}'$ is an SG.*

**Lemma 4.3** *For any $V$ in an SG $\mathcal{G}$, let $\mathcal{G}^{\overline{V}}$ be obtained from $\mathcal{G}$ by adding $W \to Z$, whenever $W \to \circ - \ldots - \circ - Z \leftarrow V$ exists in $\mathcal{G}$. Then $\mathcal{G}^{\overline{V}}$ is an SG, and $\mathcal{P}(\mathcal{G})^V = \mathcal{P}(\mathcal{G}^{\overline{V}})^V$.*

This lemma establishes that two graphs, one an edge supergraph of the other, agree on the conditional independences not involving $V$. Certainly the subgraph advertises at least as many constraints as the supergraph. To see the converse, note that definition of s-separation, coupled with our inability to condition on $V$ can always be used to create dependence between $W$ and $Z$, the vertices joined by an edge in the supergraph explicitly. This dependence can be created regardless of the conditioning set, either via the path $W \to \circ - \ldots - \circ - Z$, or via the walk path $W \to \circ - \ldots - \circ - Z \leftarrow V \to Z$. It can thus be shown that adding these edges does not remove any independences.

**Lemma 4.4** *Let $V$ be a vertex in a SG $\mathcal{G}$ with at least two vertices. Then there exists an SG $\mathcal{G}^{\underline{V}}$ where $V \to \circ - \circ$ does not exist, and $\mathcal{P}(\mathcal{G})^V = \mathcal{P}(\mathcal{G}^{\underline{V}})^V$.*

*Proof:* This follows by an inductive application of Lemmas 4.1, 4.2, and 4.3. □

Note that Lemma 4.4 does not guarantee that the graph $\mathcal{G}^{\underline{V}}$ is unique. In fact, depending on the order in which we apply the induction, we may obtain different SGs with the required property.

**Theorem 4.1** *If $\mathcal{G}$ is an SG with at least 2 vertices $\mathbf{V}$, and $V \in \mathbf{V}$, there exists an SG $\mathcal{G}^V$ with vertices $\mathbf{V} \setminus \{V\}$ such that $\mathcal{P}(\mathcal{G})^V = \mathcal{P}(\mathcal{G}^V)^V$.*

This theorem exploits previous results to construct a graph which agrees with $\mathcal{G}$ on all independences not involving $V$ and which does not contain children of $V$ that are a part of the block with size greater than two. Given a graph with this structure, we can adapt the latent projection construction to yield a SG that preserves all independences.

**Corollary 4.1** *Let $\mathcal{G}$ be an SG with vertices $\mathbf{V}$. Then for any $\mathbf{W} \subset \mathbf{V}$, there exists an SG $\mathcal{G}^*$ with vertices $\mathbf{V} \setminus \mathbf{W}$ such that $\mathcal{P}(\mathcal{G})^{\mathbf{W}} = \mathcal{P}(\mathcal{G}^*)$.*

## 5 Segregated Factorization

We now show that, for positive distributions, the Markov property we defined and a certain factorization for SGs give the same model.

A set of vertices that form a connected component in a graph obtained from $\mathcal{G}$ by dropping all edges except $\leftrightarrow$, and where no vertex is adjacent to a $-$ edge in $\mathcal{G}$ is called a *district* in $\mathcal{G}$. A *non-trivial block* is a set of vertices forming a connected component of size two or more in a graph obtained from $\mathcal{G}$ by dropping all edges except $-$. We denote the set of districts, and non-trivial blocks in $\mathcal{G}$ by $\mathcal{D}(\mathcal{G})$, and $\mathcal{B}^*(\mathcal{G})$, respectively. It is trivial to show that in a SG $\mathcal{G}$ with vertices $\mathbf{V}$, $\mathcal{D}(\mathcal{G})$, and $\mathcal{B}^*(\mathcal{G})$ partition $\mathbf{V}$.

For a vertex set $\mathbf{S}$ in $\mathcal{G}$, define $\mathrm{pa}_{\mathcal{G}}^s(\mathbf{S}) \equiv \{W \notin \mathbf{S} \mid (WV)_{\to} \text{ is in } \mathcal{G}, V \in \mathbf{S}\}$, and $\mathrm{pa}_{\mathcal{G}}^*(\mathbf{S}) \equiv \mathrm{pa}_{\mathcal{G}}^s(\mathbf{S}) \cup \mathbf{S}$. For $\mathbf{A} \subseteq \mathbf{V}$ in $\mathcal{G}$, let $\mathcal{G}_{\mathbf{A}}$ be the subgraph of $\mathcal{G}$ containing only vertices in $\mathbf{A}$ and edges between them. The *anterior* of a set $\mathbf{S}$, denoted by $\mathrm{ant}_{\mathcal{G}}(\mathbf{S})$ is the set of vertices $V$ with a partially directed path into a node in $\mathbf{S}$. A set $\mathbf{A} \subseteq \mathbf{V}$ is called *anterial* in $\mathcal{G}$ if whenever $V \in \mathbf{A}$, $\mathrm{ant}_{\mathcal{G}}(V) \subseteq \mathbf{A}$. We denote the set of non-empty anterial subsets of $\mathbf{V}$ in $\mathcal{G}$ by $\mathcal{A}(\mathcal{G})$. Let $\mathcal{D}^a(\mathcal{G}) \equiv \bigcup_{\mathbf{A} \in \mathcal{A}(\mathcal{G})} \mathcal{D}(\mathcal{G}_{\mathbf{A}})$. A *clique* in an UG $\mathcal{G}$ is a maximal connected component. The set of cliques in an UG $\mathcal{G}$ will be denoted by $\mathcal{C}(\mathcal{G})$. A vertex ordering $\prec$ is *topological* for a SG $\mathcal{G}$ if whenever $V \prec W$, $W \notin \mathrm{ant}_{\mathcal{G}}(V)$. For a vertex $V$ in $\mathcal{G}$, and a topological $\prec$, define $\mathrm{pre}_{\mathcal{G}, \prec}(V) \equiv \{W \neq V \mid W \prec V\}$.

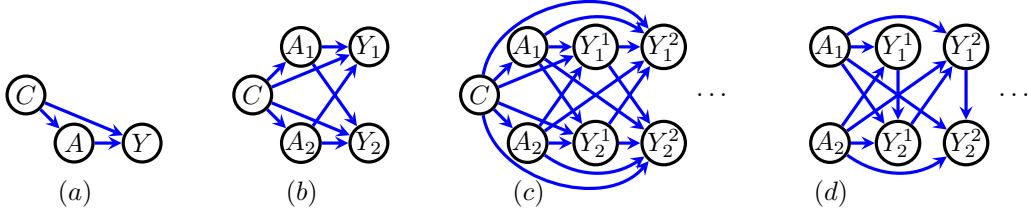

Figure 2: (a) A simple causal DAG model. (b),(c) Causal DAG models for interference. (d) A causal DAG representing a Markov chain with an equilibrium distribution in the chain graph model in Fig. 1 (a).

Given a SG $\mathcal{G}$, define the *augmented graph* $\mathcal{G}^a$ to be an undirected graph with the same vertex set as $\mathcal{G}$ where $A, B$ share an undirected edge in $\mathcal{G}^a$ if $A, B$ are connected by a walk consisting exclusively of collider sections in $\mathcal{G}$ (note that this trivially includes all $A, B$ that share an edge). We say $p(\mathbf{V})$ satisfies the augmented global Markov property with respect to a SG $\mathcal{G}$ if for any $\mathbf{A} \in \mathcal{A}(\mathcal{G})$, $p(\mathbf{A})$ satisfies the UG global Markov property with respect to $(\mathcal{G}_{\mathbf{A}})^a$. We denote a model, that is a set of $p(\mathbf{V})$ satisfying this property with respect to $\mathcal{G}$, as $\mathcal{P}^a(\mathcal{G})$.

By analogy with the ordinary Markov model and the chain graph model, we say that $p(\mathbf{V})$ obeys the *segregated factorization* with respect to a SG $\mathcal{G}$ if there exists a set of *kernels* [8] $\left\{ f_{\mathbf{S}}(\mathbf{S} \mid \mathrm{pa}_{\mathcal{G}}^s(\mathbf{S})) \big| \mathbf{S} \in \mathcal{D}^a(\mathcal{G}) \cup \mathcal{B}^*(\mathcal{G}) \right\}$ such that for every $\mathbf{A} \in \mathcal{A}(\mathcal{G})$, $p(\mathbf{A}) = \prod_{\mathbf{S} \in \mathcal{D}(\mathcal{G}_{\mathbf{A}}) \cup \mathcal{B}^*(\mathcal{G}_{\mathbf{A}})} f_{\mathbf{S}}(\mathbf{S} \mid \mathrm{pa}_{\mathcal{G}}^s(\mathbf{S}))$, and for every $\mathbf{S} \in \mathcal{B}^*(\mathcal{G})$, $f_{\mathbf{S}}(\mathbf{S} \mid \mathrm{pa}_{\mathcal{G}}^s(\mathbf{S})) = \prod_{\mathbf{C} \in \mathcal{C}((\mathcal{G}_{\mathrm{pa}_{\mathcal{G}}^*(\mathbf{S})})^a)} \phi_{\mathbf{C}}(\mathbf{C})$, where $\phi_{\mathbf{C}}(\mathbf{C})$ is a mapping from values of $\mathbf{C}$ to non-negative reals.

**Lemma 5.1** *If $p(\mathbf{V})$ factorizes with respect to $\mathcal{G}$ then $f_{\mathbf{S}}(\mathbf{S} \mid \mathrm{pa}_{\mathcal{G}}^s(\mathbf{S})) = p(\mathbf{S} \mid \mathrm{pa}_{\mathcal{G}}^s(\mathbf{S}))$ for every $\mathbf{S} \in \mathcal{B}^*(\mathcal{G})$, and $f_{\mathbf{S}}(\mathbf{S} \mid \mathrm{pa}_{\mathcal{G}}^s(\mathbf{S})) = \prod_{V \in \mathbf{S}} p(V \mid \mathrm{pre}_{\mathcal{G}, \prec}(V) \cap \mathrm{ant}_{\mathcal{G}}(\mathbf{S}))$ for every $\mathbf{S} \in \mathcal{D}^a(\mathcal{G})$ and any topological ordering $\prec$ on $\mathcal{G}$.*

**Theorem 5.1** *If $p(\mathbf{V})$ factorizes with respect to a SG $\mathcal{G}$, then $p(\mathbf{V}) \in \mathcal{P}^a(\mathcal{G})$.*

**Lemma 5.2** *If there exists a walk $\alpha$ in $\mathcal{G}$ between $A \in \mathbf{A}, B \in \mathbf{B}$ with all non-collider sections not intersecting $\mathbf{C}$, and all collider sections in $\mathrm{ant}_{\mathcal{G}}(\mathbf{A} \cup \mathbf{B} \cup \mathbf{C})$, then there exist $A^* \in \mathbf{A}, B^* \in \mathbf{B}$ such that $A^*$ and $B^*$ are s-connected given $\mathbf{C}$ in $\mathcal{G}$.*

**Theorem 5.2** $\mathcal{P}(\mathcal{G}) = \mathcal{P}^a(\mathcal{G})$.

**Theorem 5.3** *For a SG $\mathcal{G}$, if $p(\mathbf{V}) \in \mathcal{P}(\mathcal{G})$ and is positive, then $p(\mathbf{V})$ factorizes with respect to $\mathcal{G}$.*

**Corollary 5.1** *For any SG $\mathcal{G}$, if $p(\mathbf{V})$ is positive, then $p(\mathbf{V}) \in \mathcal{P}(\mathcal{G})$ if and only if $p(\mathbf{V})$ factorizes with respect to $\mathcal{G}$.*

## 6 Causal Inference and Interference Analysis

In this section we briefly describe interference analysis in causal inference, as a motivation for the use of SGs. Causal inference is concerned with using observational data to infer cause effect relationships as encoded by interventions (setting variable valus from "outside the model."). *Causal DAGs* are often used as a tool, where directed arrows represent causal relationships, not just statistical relevance. See [12] for an extensive discussion of causal inference. Much of recent work on interference in causal inference, see for instance [10, 19], has generalized causal DAG models to settings where an intervention given to a subjects affects other subjects. A classic example is herd immunity in epidemiology – vaccinating a subset of subjects can render all subjects, even those who were not vaccinated, immune. Interference is typically encoded by having vertices in a causal diagram represent not response variability in a population, but responses of individual units, or appropriately defined groups of units, where interference only occurs between groups, not within a

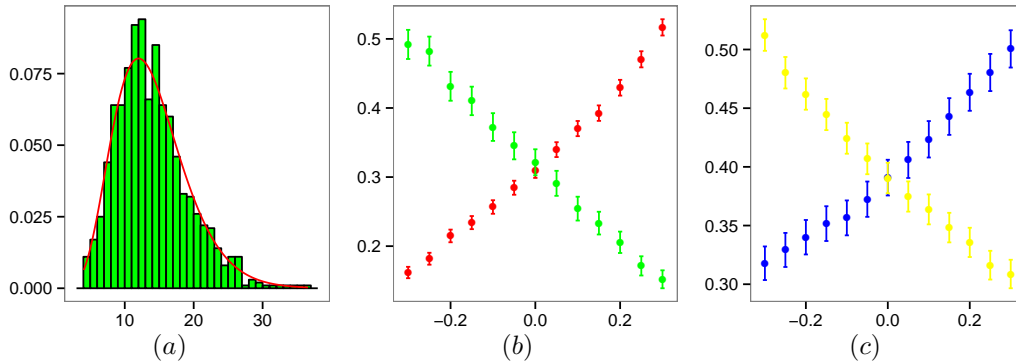

Figure 3: (a) $\chi^2$ density with 14 degrees of freedom (red) and a histogram of observed deviances of ordinary Markov models of Fig. 1 (d) fitted with data sampled from a randomly sampled model of Fig. 1 (b). (b) Y axis: values of parameters $p(Y_5 = 0 \mid Y_4 = 0, A = 0)$ (red), and $p(Y_5 = 0 \mid Y_4 = 1, A = 0)$ (green) in the fitted nested Markov model of Fig. 1 (d). X axis: value of the interaction parameter $\lambda_{45}$ (and $3 \cdot \lambda_{145}$) in the underlying chain graph model for Fig. 1. (c) Same plot with $p(Y_5 = 0 \mid Y_4 = 0, A = 1)$ (yellow), and $p(Y_5 = 0 \mid Y_4 = 1, A = 1)$ (blue).

group. For example, the DAG in Fig. 2 (b) represents a generalization of the model in Fig. 2 (a) to a setting with unit pairs where assigning a vaccine to one unit may also influence another unit, as was the case in the example in Section 2. Furthermore, we may consider more involved examples of interference if we record responses over time, as is shown in Fig. 2 (c). Extensive discussions on this type of modeling approach can be found in [18, 10].

We consider an alternative approach to encoding interference between responses using chain graph models. We give two justifications for the use of chain graphs. First, we may assume that interference arises as a dependence between responses $Y_1$ and $Y_2$ in equilibrium of a Markov chain where transition probabilities represent the causal influence of $Y_1$ on $Y_2$, and vice versa, at multiple points in time before equilibrium is reached. Under certain assumptions [9], it can be shown that such an equilibrium distribution obeys the Markov property of a chain graph. For example the DAG shown in Fig. 2 encodes transition probabilities $p(Y_1^{t+1}, Y_1^{t+1} \mid Y_1^t, Y_2^t, a_1, a_2) = p(Y_2^{t+1} \mid Y_1^{t+1}, a_1, a_2)p(Y_1^{t+1} \mid Y_2^t, a_1, a_2)$, for particular values $a_1, a_2$. For suitably chosen conditional distributions, these transition probabilities lead to an equilibrium distribution that lies in the model corresponding to the chain graph in Fig. 1 (a) [9]. Second, we may consider certain independence assumptions in our problem as reasonable, and sometimes such assumptions lead naturally to a chain graph model. For example, we may study the effect of a marketing intervention in a social network, and consider it reasonable that we can predict the response of any person only knowing the treatment for that person and responses of all friends of this person in a social network (in other words, the treatments on everyone else are irrelevant given this information). These assumptions result in a response model that is a chain graph with directed arrows from treatment to every person's response, and undirected edges between friends only.

## 6.1 An Example of Interference Analysis Using Segregated Graph Models

Given ubiquity of unobserved confounding variables in causal inference, and our our choice of chain graphs for modeling interference, we use models represented by SGs to avoid having to deal with a hidden variable chain graph model directly, due to the possibility of misspecifying the likely high dimensional hidden variables involved. We briefly describe a simulation we performed to illustrate how SGs may be used for interference analysis.

As a running example, we used a model shown in Fig. 1 (b), with $A, W, B_1, Y_1, Y_2$ binary, and $U$ 15-valued. We first considered the following family of parameterizations. In all members of this family, $A$ was assigned via a fair coin, $p(W \mid A, U)$ was a logistic model with no interactions, $B_1$ was randomly assigned via a fair coin given no complications ($W = 1$), otherwise $B_1$ was heavily weighted (0.8 probability) towards treatment assignment. The distribution $p(Y_1, Y_2 \mid U, B_1, A)$ was

obtained from a joint distribution $p(Y_1, Y_2, U, B_1, A)$ in a log-linear model of an undirected graph $\mathcal{G}$ of the form: $\frac{1}{Z} \exp \left( \sum_C (-1)^{\|x_C\|_1} \lambda_C \right)$, where $C$ ranges over all cliques in $\mathcal{G}$, $\|.\|_1$ is the $L_1$-norm, $\lambda_C$ are interactions parameters, and $Z$ is a normalizing constant. In our case $\mathcal{G}$ was an undirected graph over $A, B_1, U, Y_1, Y_2$ where edges from $Y_2$ to $B_1$ and $U$ were missing, and all other edges were present. Parameters $\lambda_C$ were generated from $\mathcal{N}(0, 0.3)$. It is not difficult to show that all elements in our family lie in the chain graph model in Fig. 1 (b).

Since all observed variables in our example are binary, the saturated model has $2^5 - 1 = 31$ parameters, and the model corresponding to Fig. 1 (d) is missing 14 of them. 2 are missing because $p(B_1 \mid W, A)$ does not depend on $A$, and 12 are missing because $p(Y_2 \mid Y_1, B_1, W, A)$ does not depend on $W, B_1$. If our results on SGs are correct, we would expect the ordinary Markov model [6] of a graph in Fig. 1 (b) to be a good fit for the data generated from our hidden variable chain graph family, where we omit the values of $U$. In particular, we would expect the observed deviances of our models fitted to data generated from our family to closely follow a $\chi^2$ distribution with 14 degrees of freedom. We generated 1000 members of our family described above, used each member to generate 5000 samples, and fitted the ordinary Markov model using an approach described in [5]. The resulting deviances, plotting against the appropriate $\chi^2$ distribution, are shown in Fig. 3 (a), which looks as we expect. We did not vary the parameters for $A, W, B_1$. This is because models for Fig. 1 (b) and Fig. 1 (d) will induce the same marginal model for $p(A, B_1, W)$ by construction.

In addition, we wanted to illustrate that we can encode interaction parameters directly via parameters in a SG. To this end, we generated a set of distributions $p(Y_1, Y_2 \mid A, U, B_1)$ via the binary log-linear model as described above, where all $\lambda_C$ parameters were fixed, except we constrained $\lambda_{\{Y_1, Y_2\}}$ to equal $3 \cdot \lambda_{\{A, Y_1, Y_5\}}$, and varied $\lambda_{\{Y_1, Y_2\}}$ from $-0.3$ to $0.3$. These parameters represent two-way interaction of $Y_1$ and $Y_2$, and three-way interaction of $A, Y_1$ and $Y_2$, and thus directly encode the strength of the interference relationship between responses. Since the SG in Fig. 1 (d) "breaks the symmetry" by replacing the undirected edge between $Y_1$ and $Y_2$ by a directed edge, the strength of interaction is represented by the degree of dependence of $Y_2$ and $Y_1$ conditional on $A$. As can be seen in Fig. 3 (b),(c) we obtain independence precisely when $\lambda_{\{Y_4, Y_5\}}$ and $\lambda_{\{A, Y_4, Y_5\}}$ in the underlying hidden variable chain graph model is 0, as expected.

Our simulations did not require the modification of the fitting procedure in [5], since Fig. 1 (d) is an ADMG. In general, a SG will have undirected blocks. However, the special property of SGs allows for a trivial modification of the fitting procedure. Since the likelihood decomposes into pieces corresponding to districts and blocks of the SG, we can simply fit each district piece using the approach in [5], and each block piece using any of the existing fitting procedures for discrete chain graph models.

## 7  Discussion and Conclusions

In this paper we considered a graphical representation of the ordinary Markov chain graph model, the set of distributions defined by conditional independences implied by a marginal of a chain graph model. We show that this model can be represented by *segregated graphs* via a global Markov property which generalizes Markov properties in chain graphs, DAGs, and mixed graphs representing marginals of DAG models. Segregated graphs have the property that bidirected and undirected edges are never adjacent. Under positivity, this global Markov property is equivalent to *segregated factorization* which decomposes the joint distribution into pieces that correspond either to sections of the graph containing bidirected edges, or sections of the graph containing undirected edges, but never both together. The convenient form of this factorization implies many existing results on chain graph and ordinary Markov models, in particular parameterizations and fitting algorithms, carry over. We illustrated the utility of segregated graphs for interference analysis in causal inference via simulated datasets.

### Acknowledgements

The author would like to thank Thomas Richardson for suggesting mixed graphs where $-$ and $\leftrightarrow$ edges do not meet as interesting objects to think about, and Elizabeth Ogburn and Eric Tchetgen Tchetgen for clarifying discussions of interference. This work was supported in part by an NIH grant R01 AI104459-01A1.

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
