[Supplementary Material · sup_cr.pdf]

# Supplementary Material to Segregated Graphs and Marginals of Chain Graph Models

**Ilya Shpitser**
Department of Computer Science
Johns Hopkins University
ilyas@cs.jhu.edu

## 1 Proofs

**Lemma 3.1.** *The Markov properties defined by superactive routes (walks) [10] in CGs, m-separation [8] in ADMGs, and d-separation [6] in DAGs are special cases of the Markov property defined by s-separation in SGs.*

*Proof:* An argument that d-separation in DAGs is a special case of the separation criterion based on superactive routes appears in [10]. An argument that d-separation in DAGs is a special case of m-separation in ADMGs trivially follows by definition. That separation based on superactive routes is a special case of s-separation follows from the fact that CGs are a special case of SGs with no $\leftrightarrow$ edges, which implies only directed edges can result in collider sections in CGs. That m-separation is a special case of s-separation follows by extension of the argument in [10]. □

**Lemma 4.1.** *For $V$ sensitive in a SG $\mathcal{G}$, let $\mathcal{G}^{\langle V \rangle}$ be the graph be obtained from $\mathcal{G}$ by replacing all $-$ edges adjacent to $V$ by $\rightarrow$ edges pointing away from $V$. Then $\mathcal{G}^{\langle V \rangle}$ is an SG, and $\mathcal{P}(\mathcal{G}) = \mathcal{P}(\mathcal{G}^{\langle V \rangle})$.*

*Proof:* Since $\mathcal{G}^{\langle V \rangle}$ is constructed from an SG by replacing certain $-$ edges by $\rightarrow$, then if $\mathcal{G}$ does not contain $\circ \leftrightarrow \circ - \circ$, then neither does $\mathcal{G}^{\langle V \rangle}$. If $\mathcal{G}^{\langle V \rangle}$ contains a partially directed cycle not including $V$, so does $\mathcal{G}$, which is a contradiction. If $\mathcal{G}^{\langle V \rangle}$ contains a partially directed cycle including $V$, then it must be via a subpath $\circ \rightarrow V \rightarrow \circ$, with all other edges on the path present in $\mathcal{G}$. But either the outgoing edge from $V$ that is on the cycle is also present in $\mathcal{G}$ or it is undirected. In both cases, there is still a partially directed cycle in $\mathcal{G}$, which is a contradiction. Thus $\mathcal{G}^{\langle V \rangle}$ is a SG. If $V$ has no adjacent $-$ edges, $\mathcal{G} = \mathcal{G}^{\langle V \rangle}$.

Assume $(\mathbf{A} \not\perp\!\!\!\perp \mathbf{B} \mid \mathbf{C})_{\mathcal{G}^{\langle V \rangle}}$. Fix a walk $\alpha$ from $\mathbf{A}$ to $\mathbf{B}$ s-connected by $\mathbf{C}$ in $\mathcal{G}^{\langle V \rangle}$. We will construct an s-connected walk $\alpha^*$ from $\mathbf{A}$ to $\mathbf{B}$ given $\mathbf{C}$ in $\mathcal{G}$. By definition, every collider section in $\alpha$ intersects $\mathbf{C}$ and every non-collider section in $\alpha$ is free of $\mathbf{C}$. Any section of $\alpha$ where $V$ does not occur either remains a section of $\alpha$ in $\mathcal{G}$, and retains its open status (if its neighboring edges do not change status in $\mathcal{G}$), or is subsumed by the argument for the following case. We now consider all sections $\beta_i$ of $\alpha$ where $V$ occurs. Note that $\beta_i$ is a singleton section. If $\beta_i$ is a collider section, $V \in \mathbf{C}$, and $\beta_i$ exists in $\mathcal{G}$. Assume $\beta_i$ is a non-collider section. Then $V \notin \mathbf{C}$. If $\beta_i$ is in $\mathcal{G}$, we are done. Otherwise, consider a section $\beta_j$ in $\alpha^*$ containing sections $\beta_{i-l}, \ldots, \beta_{i+k}$ in $\alpha$. By definition of $\mathcal{G}^{\langle V \rangle}$, all sections except possibly $\beta_{i-l}$ and $\beta_{i+k}$ are either of the form $\leftarrow V \rightarrow$ or collider sections. Note that since $\alpha$ is open, all collider sections intersect $\mathbf{C}$.

If $\beta_j$ is a collider section, we are done. Otherwise, we have two cases. If both neighboring edges along $\alpha^*$ into $\beta_j$ are not into $\beta_j$, then $\beta_i \leftarrow V \rightarrow \beta_{i+k}$ shares the same endpoint behavior as $\beta_j$ and is open, since $\beta_i, \leftarrow V \rightarrow$, and $\beta_{i+k}$ are non-collider sections in $\alpha$ and thus do not intersect $\mathbf{C}$. If a single neighboring edge along $\alpha^*$ into $\beta_j$ is into $\beta_j$ (say into $\beta_{i-l}$), then either that edge is from $V$ or not. If it is from $V$, the section $V \rightarrow \beta_{i+k}$ shares the same endpoint behavior as $\beta_j$ and is open. If it is not from $V$, but another edge $W$, then since $V$ is sensitive, $W \rightarrow V$ exists in $\mathcal{G}$, and the section $W \rightarrow V \rightarrow \beta_{i+k}$ shares the same endpoint behavior as $\beta_j$ and is open.

Assume $(\mathbf{A} \not\perp\!\!\!\perp \mathbf{B} \mid \mathbf{C})_{\mathcal{G}}$. Fix a walk $\alpha$ from $\mathbf{A}$ to $\mathbf{B}$ s-connected by $\mathbf{C}$ in $\mathcal{G}$. We will construct an s-connected walk from $\mathbf{A}$ to $\mathbf{B}$ given $\mathbf{C}$ in $\mathcal{G}^{\langle V \rangle}$. By definition, every collider section in $\alpha$ intersects $\mathbf{C}$ and every non-collider section in $\alpha$ is free of $\mathbf{C}$. Any section of $\alpha$ where $V$ does not occur remains a section of $\alpha$ in $\mathcal{G}^{\langle V \rangle}$, and retains its open status. We now consider all sections $\beta_i$ of $\alpha$ where $V$ occurs.

Assume $\beta_i$ is a collider section with end points $Z, W$. If $V \in \mathbf{C}$, then since $V$ is sensitive, $Z \to V \leftarrow W$ is present in $\mathcal{G}^{\langle V \rangle}$. Then we can construct a walk $\alpha'$ which shares all sections with $\alpha$ except $\beta_i$ is replaced by $Z \to V \leftarrow W$, which is open since $V \in \mathbf{C}$. If $V \notin \mathbf{C}$, then there must be some section $\beta_j$ in $\beta_i$ in $\mathcal{G}^{\langle V \rangle}$ intersecting $\mathbf{C}$. This section either has $V$ as both endpoints, or $V$ and an endpoint $Z$ of $\beta_i$ with an arrowhead into $\beta_i$. We can then replace $\alpha$ with another walk $\alpha'$ which shares all sections with $\alpha$ except $\beta_i$ is replaced either by $W \to V \beta_j V \leftarrow Z$, or $W \to V \beta_j \leftarrow Z$, which is open since $\beta_j$ intersects $\mathbf{C}$. In either case, we then repeat the argument for other sections of $\alpha'$.

Assume $\beta_i$ is a non-collider section with end points $Z, W$, and does not intersect $\mathbf{C}$. This means there is at most one arrowhead into $\beta_i$, say from $Z$, or no arrowheads into $\beta_i$. In the former case, fix the section $\beta_j$ (possibly of length 0 if $V = W$) in $\mathcal{G}^{\langle V \rangle}$ between the last occurrence of $V$ and $W$ in $\beta_i$. Replace $\alpha$ by a walk $\alpha'$ sharing all sections with $\alpha$ except $\beta_i$ is replaced with $Z \to V \beta_j W$, which is open. If no arrowheads are into $\beta_i$, let $\beta_j$ be the part of $\beta_i$ from $Z$ to first occurrence of $V$, and $\beta_k$ be the part of $\beta_i$ from the last occurrence of $V$ to $W$. Replace $\alpha$ by a walk $\alpha'$ sharing all sections with $\alpha$ except $\beta_i$ is replaced by $Z \beta_j V \beta_k W$. In all cases, the newly added sections to $\alpha'$ are open and share end edge behavior with sections they are replacing. We then repeat the argument for other sections of $\alpha'$. Thus, $(\mathbf{A} \not\perp\!\!\!\perp \mathbf{B} \mid \mathbf{C})_{\mathcal{G}^{\langle V \rangle}}$. $\qquad\square$

**Lemma 4.2.** *Let $\mathcal{G}$ be an SG, and $\mathcal{G}'$ a graph obtained from adding an edge $W \to V$ for two non-adjacent vertices $W, V$ where $W \to \circ - \ldots - \circ - V$ exists in $\mathcal{G}$. Then $\mathcal{G}'$ is an SG.*

*Proof:* Since $\mathcal{G}$ is an SG, and we are adding only $\to$ edges to $\mathcal{G}'$, then there is no $\circ \leftrightarrow \circ - \circ$ structure in $\mathcal{G}'$. If there were a partially directed cycle involving $W \to V$ in $\mathcal{G}'$, then replacing $W \to V$ by $W \to \circ - \ldots - \circ - V$ in the cycle would still result in a partially directed cycle, which would also be present in $\mathcal{G}$. But this is a contradiction. $\qquad\square$

**Lemma 4.3.** *For any $V$ in an SG $\mathcal{G}$, let $\mathcal{G}^{\overline{V}}$ be obtained from $\mathcal{G}$ by adding $W \to Z$, whenever $W \to \circ - \ldots - \circ - Z \leftarrow V$ exists in $\mathcal{G}$. Then $\mathcal{G}^{\overline{V}}$ is an SG, and $\mathcal{P}(\mathcal{G})^V = \mathcal{P}(\mathcal{G}^{\overline{V}})^V$.*

*Proof:* $\mathcal{G}^{\overline{V}}$ is an SG by an inductive application of Lemma 4.2. If $\mathbf{A} \perp\!\!\!\perp \mathbf{B} \mid \mathbf{C}$ holds in $\mathcal{G}^{\overline{V}}$, then $\mathbf{A} \perp\!\!\!\perp \mathbf{B} \mid \mathbf{C}$ holds in $\mathcal{G}$, since $\mathcal{G}$ is an edge subgraph of $\mathcal{G}^{\overline{V}}$.

Assume $(\mathbf{A} \perp\!\!\!\perp \mathbf{B} \mid \mathbf{C})_{\mathcal{G}}$, where $V \notin \mathbf{A} \cup \mathbf{B} \cup \mathbf{C}$. Fix a walk $\alpha$ from $\mathbf{A}$ to $\mathbf{B}$ in $\mathcal{G}^{\overline{V}}$. If $\alpha$ exists in $\mathcal{G}$, then it retains the same edges in $\mathcal{G}^{\overline{V}}$, which implies if $\alpha$ is s-separated by $\mathbf{S}$ in $\mathcal{G}$, it is also in $\mathcal{G}^{\overline{V}}$. Assume $\alpha$ does not exist in $\mathcal{G}$ and is s-connected given $\mathbf{C}$. This means $\alpha$ contains a set of edges of the form $W \to Z$ which do not exist in $\mathcal{G}$. We will repeatedly replace edges $W \to Z$ in $\alpha$ by sections that exist in $\mathcal{G}$ while preserving the open status of the resulting walk. In this way, we will construct a new walk that is s-connected given $\mathbf{C}$ and exists in $\mathcal{G}$, deriving a contradiction.

Pick an edge $W \to Z$ in $\alpha$ that does not exist in $\mathcal{G}$, let $\beta_j$ be the section of $\alpha$ starting at $Z$ with $W \to Z$ pointing into it. By definition of $\mathcal{G}^{\overline{V}}$, there exists $\beta_i \equiv W \to \circ - \ldots - \circ - Z$ in $\mathcal{G}$. If $\beta_j$ is a collider section, then replace $W \to \beta_j$ by $\beta_i \beta_j$. The new extended section is thus also a collider section intersecting $\mathbf{C}$, and exists in $\mathcal{G}$. If $\beta_j$ is not a collider section, then either $\beta_i$ intersects $\mathbf{C}$ or not. If it does, replace $W \to \beta_j$ by $\beta_i \leftarrow V \to \beta_j$. This results in three new sections which are all open given $\mathbf{C}$, exist in $\mathcal{G}$, and have same endpoint behavior as $\beta_j$. If it does not, replace $W \to \beta_j$ by $\beta_i \beta_j$. This results in a new extended section which is a non-collider section that does not intersect $\mathbf{C}$, exists in $\mathcal{G}$, and has same endpoint behavior as $\beta_j$.

Repeating the argument for every $W \to V$ that does not exist in $\mathcal{G}$ gives us the contradiction. $\qquad\square$

**Theorem 4.1.** *If $\mathcal{G}$ is an SG with at least 2 vertices $\mathbf{V}$, and $V \in \mathbf{V}$, there exists an SG $\mathcal{G}^V$ with vertices $\mathbf{V} \setminus \{V\}$ such that $\mathcal{P}(\mathcal{G})^V = \mathcal{P}(\mathcal{G}^V)^V$.*

*Proof:* Construct $\mathcal{G}^{\underline{V}}$ as in Lemma 4.4. Construct $\mathcal{G}^V$ from $\mathcal{G}^{\underline{V}}$ as follows. Retain all vertices in $\mathbf{V} \setminus \{V\}$ and edges between them. For any two vertices $W, Z$: if $W \to V \to Z$, add $W \to Z$; if

$W \leftarrow V \rightarrow Z$, add $W \leftrightarrow Z$; if $W - V - Z$, add $W - Z$; if $W - V \rightarrow Z$, add $W \rightarrow Z$; and if $W \rightarrow V - Z$, add $W \rightarrow Z$.

Because $\mathcal{G}^{\underline{V}}$ is a SG with no $V \rightarrow \circ - \circ$, there is no $\circ \leftrightarrow \circ - \circ$ structure in $\mathcal{G}^{V}$. Assume there exists a partially directed cycle in $\mathcal{G}^{V}$ involving new edges. Then we can systematically replace them by the two edge paths in $\mathcal{G}$ to yield a partially directed cycle in $\mathcal{G}$, giving a contradiction.

Let $\mathcal{G}^{V^{\dagger}}$ be an edge supergraph of $\mathcal{G}^{\underline{V}}$ where we add all edges in $\mathcal{G}^{V}$ that do not exist in $\mathcal{G}$. We first show $\mathcal{P}(\mathcal{G}^{V^{\dagger}})^{V} = \mathcal{P}(\mathcal{G}^{\underline{V}})^{V}$. If $(\mathbf{A} \perp\!\!\!\perp \mathbf{B} \mid \mathbf{C})_{\mathcal{G}^{V^{\dagger}}}$, then $(\mathbf{A} \perp\!\!\!\perp \mathbf{B} \mid \mathbf{C})_{\mathcal{G}^{\underline{V}}}$ because $\mathcal{G}^{V^{\dagger}}$ is an edge supergraph of $\mathcal{G}^{\underline{V}}$. Assume $(\mathbf{A} \perp\!\!\!\perp \mathbf{B} \mid \mathbf{C})_{\mathcal{G}^{\underline{V}}}$, and fix a walk $\alpha$ from $\mathbf{A}$ to $\mathbf{B}$ that is s-connected given $\mathbf{C}$ in $\mathcal{G}^{V^{\dagger}}$. If $\alpha$ exists in $\mathcal{G}^{\underline{V}}$, we have a contradiction. Otherwise, since $V \notin \mathbf{A} \cup \mathbf{B} \cup \mathbf{C}$, it is easy to construct a walk $\alpha'$ that is s-connected given $\mathbf{C}$ and exists in $\mathcal{G}^{\underline{V}}$ by replacing edges in $\alpha$ that do not exist in $\mathcal{G}^{\underline{V}}$ by their corresponding two edges used in the construction of $\mathcal{G}^{V}$.

Finally, we show that $\mathcal{P}(\mathcal{G}^{V^{\dagger}})^{V} = \mathcal{P}(\mathcal{G}^{V})^{V}$. Since $\mathcal{G}^{V^{\dagger}}$ is an edge supergraph of $\mathcal{G}^{V}$, if $\mathbf{A} \perp\!\!\!\perp \mathbf{B} \mid \mathbf{C}$ in $\mathcal{G}^{V^{\dagger}}$, then $\mathbf{A} \perp\!\!\!\perp \mathbf{B} \mid \mathbf{C}$ in $\mathcal{G}^{V}$. If $\mathbf{A} \perp\!\!\!\perp \mathbf{B} \mid \mathbf{C}$ in $\mathcal{G}^{V}$, and there is a s-connecting walk $\alpha$ from $\mathbf{A}$ to $\mathbf{B}$ given $\mathbf{C}$ in $\mathcal{G}^{V^{\dagger}}$, it must involve $V$. But we can construct a walk $\alpha'$ that does not contain $V$ by replacing $V$ containing segments by edges connecting nodes adjacent to $V$ following above rules used to construct $\mathcal{G}^{V}$. It is easy to see $\alpha'$ is s-connecting given $\mathbf{C}$ if $\alpha$ is. This is a contradiction. □

**Corollary 4.1.** *Let $\mathcal{G}$ be an SG with vertices $\mathbf{V}$. Then for any $\mathbf{W} \subset \mathbf{V}$, there exists an SG $\mathcal{G}^{*}$ with vertices $\mathbf{V} \setminus \mathbf{W}$ such that $\mathcal{P}(\mathcal{G})^{\mathbf{W}} = \mathcal{P}(\mathcal{G}^{*})$.*

*Proof:* Follows by an inductive application of Theorem 4.1 for any ordering of vertices in $\mathbf{W}$. □

**Lemma 5.1.** *If $p(\mathbf{V})$ factorizes with respect to $\mathcal{G}$ then $f_{\mathbf{S}}(\mathbf{S} \mid \mathrm{pa}_{\mathcal{G}}^{s}(\mathbf{S})) = p(\mathbf{S} \mid \mathrm{pa}_{\mathcal{G}}^{s}(\mathbf{S}))$ for every $\mathbf{S} \in \mathcal{B}^{*}(\mathcal{G})$, and $f_{\mathbf{S}}(\mathbf{S} \mid \mathrm{pa}_{\mathcal{G}}^{s}(\mathbf{S})) = \prod_{V \in \mathbf{S}} p(V \mid \mathrm{pre}_{\mathcal{G},\prec}(V) \cap \mathrm{ant}_{\mathcal{G}}(\mathbf{S}))$ for every $\mathbf{S} \in \mathcal{D}^{a}(\mathcal{G})$ and any topological ordering $\prec$ on $\mathcal{G}$.*

*Proof:* We will proceed by induction on anterial subgraphs. We will add either a singleton vertex that will be become a new singleton district or a part of an existing district, or a block of vertices $\mathbf{S}$ to construct $\mathcal{G}$ with vertices $\mathbf{V}$, such that $\mathbf{V} \setminus \mathbf{S} \in \mathcal{A}(\mathcal{G})$. For the base case, the conclusion clearly holds for $\mathcal{G}$ with a single vertex. Assume the inductive hypothesis holds for $\mathcal{G}^{i}$, and we added a block $\mathbf{S}$ to $\mathcal{G}^{i}$ to yield $\mathcal{G}$, where $\mathbf{S} \in \mathcal{B}^{*}(\mathcal{G})$. By the inductive hypothesis, $p(\mathbf{V}) = f_{\mathbf{S}}(\mathbf{S} \mid \mathrm{pa}_{\mathcal{G}}^{s}(\mathbf{S})) \cdot \prod_{\mathbf{S} \in \mathcal{D}(\mathcal{G}^{i}) \cup \mathcal{B}^{*}(\mathcal{G}^{i})} p(\mathbf{S} \mid \mathrm{pa}_{\mathcal{G}^{i}}^{s}(\mathbf{S}))$. This implies our conclusion. Assume the inductive hypothesis holds for $\mathcal{G}^{i}$, and we added $V$ to $\mathcal{G}^{i}$ to yield $\mathcal{G}$, where $V \in \mathbf{S} \in \mathcal{D}(\mathcal{G})$. Then the conclusion follows by a simple extension of the argument used to prove Lemma 1 in [11]. □

**Theorem 5.1.** *If $p(\mathbf{V})$ factorizes with respect to a SG $\mathcal{G}$, then $p(\mathbf{V}) \in \mathcal{P}^{a}(\mathcal{G})$.*

*Proof:* Implied by the fact that the UG factorization implies the UG global Markov property [5]. □

**Lemma 5.2.** *If there exists a walk $\alpha$ in $\mathcal{G}$ between $A \in \mathbf{A}, B \in \mathbf{B}$ with all non-collider sections not intersecting $\mathbf{C}$, and all collider sections in $\mathrm{ant}_{\mathcal{G}}(\mathbf{A} \cup \mathbf{B} \cup \mathbf{C})$, then there exist $A^{*} \in \mathbf{A}, B^{*} \in \mathbf{B}$ such that $A^{*}$ and $B^{*}$ are s-connected given $\mathbf{C}$ in $\mathcal{G}$.* [1]

*Proof:* Let $D$ be the last vertex on $\alpha$ in $\mathrm{ant}_{\mathcal{G}}(\mathbf{A}) \setminus \mathrm{ant}_{\mathcal{G}}(\mathbf{C})$ if such a vertex exists, or $D \equiv A$ otherwise. Let $E$ be the first vertex in $\mathrm{ant}_{\mathcal{G}}(\mathbf{B}) \setminus \mathrm{ant}_{\mathcal{G}}(\mathbf{C})$ which occurs between the last occurrence of $D$ in $\alpha$ and $B$, if such a vertex exists, or $E \equiv B$ otherwise. If $D \neq A$, let $A^{*}$ be any vertex such that $D \in \mathrm{ant}_{\mathcal{G}}(A^{*})$, otherwise let $A^{*} \equiv A$. Similarly, if $E \neq B$, let $B^{*}$ be any vertex such that $E \in \mathrm{ant}_{\mathcal{G}}(B^{*})$, otherwise let $B^{*} \equiv B$.

Let $\alpha^{*}$ be the subwalk of $\alpha$ between the last occurrence of $D$ and the first occurrence of $E$. Then: (a) every vertex in $\alpha^{*}$ is in $\mathrm{ant}_{\mathcal{G}}(\mathbf{C})$; (b) there is a partially directed path $\delta$ from $D$ to $A^{*}$, and $\epsilon$ from $E$ to $B^{*}$; (c) other than possibly $D$ or $E$, no vertex in $\delta$ or $\epsilon$ is in $\mathrm{ant}_{\mathcal{G}}(\mathbf{C})$; and (d) no vertex in $\epsilon$ other than possibly $E$ is an ancestor of $A^{*}$.

It follows from (a) and (c) that $\alpha^{*}$ and $\epsilon$ only intersect at $E$, and $\alpha^{*}$ and $\delta$ only intersect at $D$. Let $\beta$ be a walk obtained by concatenating $\delta$, $\alpha^{*}$, and $\gamma$. By construction, every collider section in $\alpha^{*}$ is in $\mathrm{ant}_{\mathcal{G}}(\mathbf{C})$, every non-collider section in $\alpha^{*}$ does not intersect $\mathbf{C}$. Furthermore, every section in $\delta$ and $\epsilon$ is non-collider and does not intersect $\mathbf{C}$. Thus $\beta$ is s-connecting given $\mathbf{C}$. □

**Theorem 5.2.** $\mathcal{P}(\mathcal{G}) = \mathcal{P}^a(\mathcal{G})$. [2]

*Proof:* Fix disjoint $\mathbf{A}, \mathbf{B}, \mathbf{C}$, and consider the smallest anterial set $\mathbf{A}^\dagger$ containing $\mathbf{A}, \mathbf{B}, \mathbf{C}$. By definition of s-separation, it suffices to restrict our attention to walks contained in $\mathbf{A}^\dagger$. Fix a walk $\alpha$ from $A \in \mathbf{A}$ to $B \in \mathbf{B}$ open in $\mathcal{G}_{\mathbf{A}^\dagger}$ given $\mathbf{C}$. We will construct a path $\beta$ from $A$ to $B$ in $(\mathcal{G}_{\mathbf{A}^\dagger})^a$ which does not intersect $\mathbf{C}$. Since $\alpha$ is open, every section $\alpha_1, \ldots, \alpha_k$ in $\alpha$ is open. We will first construct a walk $\alpha^\dagger$ in $(\mathcal{G}_{\mathbf{A}^\dagger})^a$ consisting of fragments corresponding to sections in $\alpha_i$, and then simplify this walk to a path that does not intersect $\mathbf{C}$. If $\alpha_i$ is a non-collider section, let $\alpha_i^\dagger$ consist of the undirected edges corresponding to those in $\alpha_i$. If $\alpha_i$ is a collider section with end points $C, D$, let $\alpha_i^\dagger$ consist of $C - D$. Then the starting vertex of $\alpha_1^\dagger$ is $A$, the ending vertex of $\alpha_k^\dagger$ is $B$, $\alpha_1^\dagger, \ldots, \alpha_k^\dagger$ are undirected walks that do not intersect $\mathbf{C}$ by construction, and for each $i \in 1, \ldots, k-1$ either $\alpha_i^\dagger$ shares the ending vertex with the starting vertex of $\alpha_{i+1}^\dagger$, or the ending vertex of $\alpha_i^\dagger$ and the starting vertex of $\alpha_{i+1}^\dagger$ are neighbors. Thus, we can construct a walk from these walks with a starting vertex $A$, ending vertex $B$, and which does not intersect $\mathbf{C}$. But this means we can construct a path $\beta$ with the same property.

Fix a minimal path $\beta$ from $A \in \mathbf{B}$ to $B \in \mathbf{B}$ that does not intersect $\mathbf{C}$ in $(\mathcal{G}_{\mathbf{A}^\dagger})^a$. We will construct a walk $\alpha$ from $A$ to $B$ s-connected given $\mathbf{C}$ in $\mathcal{G}_{\mathbf{A}^\dagger}$. Let the edges of $\beta$ be $b_1, \ldots b_k$. We will construct $\alpha$ by replacing all $b_i$ that do not exist in $\mathcal{G}_{\mathbf{A}^\dagger}$ by a witnessing collider walk, and all other $b_i$ between $C, D$ by the (possibly directed or bidirected) edge between $C, D$ in $\mathcal{G}_{\mathbf{A}^\dagger}$. The result is clearly a walk. Furthermore, all non-collider sections on this walk do not intersect $\mathbf{C}$, and all collider sections are in $\mathbf{A}^\dagger$, so in the anterior of $\mathbf{A} \cup \mathbf{B} \cup \mathbf{C}$. By lemma 5.2, there exists a walk from $A$ to $B$ s-connected given $\mathbf{C}$ in $\mathbf{A}^\dagger$. $\qquad\square$

**Theorem 5.3.** *For a SG $\mathcal{G}$, if $p(\mathbf{V}) \in \mathcal{P}(\mathcal{G})$ and is positive, then $p(\mathbf{V})$ factorizes with respect to $\mathcal{G}$.*

*Proof:* Fix any $\mathbf{D} \in \mathcal{A}(\mathcal{G})$, and a topological ordering $\prec$. By the chain rule of probabilities, $p(\mathbf{D}) = \prod_{V \in \mathbf{D}} p(V \mid \mathrm{pre}_{\mathcal{G}, \prec}(V) \cap \mathbf{D})$ which is equal to $\prod_{\mathbf{S} \in \mathcal{D}^a(\mathcal{G}_{\mathbf{D}}) \cup \mathcal{B}^*(\mathcal{G}_{\mathbf{D}})} \prod_{V \in \mathbf{S}} p(V \mid \mathrm{pre}_{\mathcal{G}, \prec}(V) \cap \mathbf{D})$ since non-trivial blocks and districts partition $\mathbf{V}$. This in turn is equal to $\prod_{\mathbf{S} \in \mathcal{D}^a(\mathcal{G}_{\mathbf{D}}) \cup \mathcal{B}^*(\mathcal{G}_{\mathbf{D}})} \prod_{V \in \mathbf{S}} p(V \mid \mathrm{pre}_{\mathcal{G}, \prec}(V) \cap \mathrm{pa}^*_{\mathcal{G}}(\mathbf{S}))$, by assumption. This implies that we obtain the outer level factorization: $p(\mathbf{D}) = \prod_{\mathbf{S} \in \mathcal{D}^a(\mathcal{G}_{\mathbf{D}}) \cup \mathcal{B}^*(\mathcal{G}_{\mathbf{D}})} f_{\mathbf{S}}(\mathbf{S} \mid \mathrm{pa}^s_{\mathcal{G}}(\mathbf{S}))$. That the inner factorization holds for any $f_{\mathbf{S}}(\mathbf{S} \mid \mathrm{pa}^s_{\mathcal{G}}(\mathbf{S}))$ for $\mathbf{S} \in \mathcal{B}^*(\mathcal{G})$ for a positive $p(\mathbf{V})$ follows from Theorem 3.36 in [5] (and ultimately the Hammersley Clifford theorem for UG models). $\qquad\square$

## Footnotes

[1] The proof follows the proof of lemma 1 in [8].

[2] This proof follows lemma 3 in [8].

[12] T. J. VanderWeele, E. J. T. Tchetgen, and M. E. Halloran. Components of the indirect effect in vaccine trials: identification of contagion and infectiousness effects. *Epidemiology*, 23(5):751–761, 2012.

[13] T. S. Verma and J. Pearl. Equivalence and synthesis of causal models. Technical Report R-150, Department of Computer Science, University of California, Los Angeles, 1990.