[Reviews · NeurIPS 2015]

Submitted by Assigned_Reviewer_1

The authors try to tackle the problem when bidirected edge (<->) and undirected edge (-) meet in a loopless mixed graph. They propose to construct a new mixed graph called segregated graph (SG) (where bidirected edge (<->) and undirected edge (-) never meet) which preserves all conditional independencies on the observed variables.

I like the originality of the idea. I am somewhat disappointed by the clarity of the paper, which affects the quality too. Also the significance of the problem considered is not well addressed. Here I give more details on each criterion: *Clarity: I like the authors' explanation of the big picture before going to details, which helps to clarify to some degree. However, once we check the details, it becomes less clear caused by typos and undefined/undeclared symbols. It would be much easier to read if the authors follow notations in any of the well known (and recent) publications on this topic such as [4], [1*], and [2*].

*Quality: it is dragged down by the level of clarity. Also I notice that if we move most of the proofs to the supplementary file, the paper would have much more spare space (2-3 pages perhaps). This suggests much more content can be added to the paper. *Originality: good. *Significance: the importance of the problem is only lightly touched (line 128-129) with a bit further discussion on the implication in conclusions.

Missing references: [1*] Sadeghi, K. and Lauritzen, S. (2014). Markov properties for mixed graphs. Bernoulli 20, 676-696. MR3178514 [2*] Wermuth, N. (2011). Probability distributions with summary graph structure. Bernoulli 17, 845-879. MR2817608

After reviewers' dissuasion and seeing authors' response, I've increased the score by one step.
Summary: This paper proposes segregated graphs in order to tackle a problem in loopless mixed graphs, where bidirected edge (<->) and undirected edge (-) meet.

Submitted by Assigned_Reviewer_2

SGs are graphical models over observed variables, consisting of undirected, directed, and bidirected edges. Bidirected edges represent a common latent parent, while undirected edges capture a bidirectional causal relationship between variables, where A can cause B or vice versa. The overall model seems to be a combination of ADMGs and MRFs.

SGs contain no partially directed cycles; also -- and <--> edges are not allowed to meet. These properties imply a factorization of the joint distribution that is a combination of ADMGs and MRF factors. If -- and <--> edges do meet in the original model, the authors show how to orient -- edges while preserving the conditional independencies; this is one of the main technical contributions.

Quality: The paper is technically sound, and all claims are supported by rigorous proofs. There are no experiments, but the authors give examples of causal interactions for which SGs would be appropriate.

Clarity: The paper is well-written; however it is very dense and some of the proofs are hard to follow. I had to draw diagrams on the margins in order to understand claims throughout the paper. The presentation would probably be clearer if the paper included more illustrative figures.

Originality: SGs were constructed by merging existing models, and as such are not that original (even though the work is highly non-trivial).

Significance: Not very significant on its own, but might have impact on future work. SGs would be more interesting if accompanied by e.g. some new results on causal identifiability (that cannot be expressed using existing models), or an inference algorithm for some parametric family.
Summary: This is a high quality paper that describes segregated graphs (SGs), a novel graphical model over observed variables, consisting of bidirected, directed, and undirected edges. The model seems to combine ADMGs and MRS, and has a bit more flexibility in specifying causal relationships.

Submitted by Assigned_Reviewer_3

The paper proposes a reformulation of a large class of mixed graphical models that allows a series of factorization. The paper is quite technical and my main concern is that I did not really to what extant the proposed representation can simplifying some problems or reduce the complexity of some inference algorithm.

I fear that the 8-page NIPS format does not fit for this kind of work, which would need a series of illustration to examplify the progress it allows.

Summary: See below

Submitted by Assigned_Reviewer_4

Hidden variables in Bayesian networks and in chain graphs add extra complexity to parameter learning and to inference. Mixed graphical models introduce bi-directed edges that can eliminate the hidden variables while preserving the conditional independencies induced by the DAG or chain graph representations. However, after applying latent projection operation to eliminate the hidden variable in the resulting mixed graphs, bi-directional and undirected edges can meet, which leads to issues when trying to factorize the model.

As a solution segregated graphs are proposed, which preserve the conditional independencies between the non-latent random variables in the original DAG or chain graph, and at the same time they avoid having nodes where undirected and bi-directed edges meet.

Clarity: The introduction uses notations that are only introduced in the background section, which makes the reading of the paper more difficult.

Significance: It is hard to judge the paper's significance without having a real world practical example and its evaluation at hand.

Typo: line 099 "...B_1,Y_1..." -> "B_1,Y_2"
Summary: Segregated graphs are proposed, which can preserve conditional independencies between non-latent random variables in a Bayesian network or chain graph, while eliminating latent variables without introducing nodes where undirected and bi-directed edges meet. This can have a significant impact, however, this is not demonstrated through a practical example and with experimental evaluation.

Author Feedback
Author rebuttal: We thank the reviewers for their thoughtful comments about the paper. Specific replies, where warranted, follow.

Reviewer 1: We wanted to thank the reviewer for the additional relevant references. We will do a better job of proofreading the paper. Because the arguments were not straightforward, we felt it was worthwhile to sacrifice some exposition to make sure all arguments were exposed to peer review.

We believe moving simpler arguments to the supplement would create space to better justify the result, and otherwise improve the clarity of the paper.

Reviewer 2: A fitting algorithm for discrete statespace models represented by segregated graphs follows immediately from our results. The likelihood is decomposable by blocks and districts, thus we can simply use the algorithm from Evans and Richardson for districts, and a chain graph fitting algorithm (e.g.
iterative proportional fitting) for block pieces to fit our model. Variants of belief propagation are possible by e.g. constructing a join tree from the augmented (moralized) graph obtained from a segregated graph in section 5. We did not discuss these extensions in detail in the paper in the interests of space -- we felt it was important to give our arguments in the main body of the paper as the main contribution is theoretical.

We agree that thinking about identification in causal SGs is very interesting. But we believe it is first necessary to define and justify counterfactuals that do not come from a directed acyclic graph causal model. This is possible, and we are pursuing this, but we feel it would be out of scope for this paper. Tchetgen Tchetgen and Vanderweele did some preliminary work on this in the context of interference in causal inference (we modified one of their examples to apply to two time points in section 2).

In terms of significance, the relationship of our work to latent variable chain graphs is similar to the relationship of Evans and Richardson's work on the ordinary Markov model in Annals, and hidden variable DAG models. It follows from our results (but this is out of scope for a conference paper) that we get a well-defined conditional independence supermodel (e.g. forms a curved exponential family), just as E&R get for their case. We believe this is important to show. We will be happy to clarify this in the text.

Reviewers 3, 5 and 6:

This paper shows that there exist "nice models" that capture much of the structure of latent variable chain graphs, a very general class of hidden variable model. Given that we wanted to prioritize exposing our arguments to peer review, showing this took most of the paper. We believe this result is important for causal inference, analysis of social networks, and other types of hidden variable modeling where directed and symmetric relationships appear together. We simply did not have the space to pursue extensions and applications further in a conference paper. We would be happy to move some of the simpler arguments to a supplement, and use the space to better justify the result, and improve the clarity of the paper.